# Learning Fairness in Multi-Agent Systems

**Jiechuan Jiang**
Peking University
jiechuan.jiang@pku.edu.cn

**Zongqing Lu**[*]
Peking University
zongqing.lu@pku.edu.cn

## Abstract

Fairness is essential for human society, contributing to stability and productivity. Similarly, fairness is also the key for many multi-agent systems. Taking fairness into multi-agent learning could help multi-agent systems become both efficient and stable. However, learning efficiency and fairness simultaneously is a complex, multi-objective, joint-policy optimization. To tackle these difficulties, we propose FEN, a novel hierarchical reinforcement learning model. We first decompose fairness for each agent and propose *fair-efficient* reward that each agent learns its own policy to optimize. To avoid multi-objective conflict, we design a hierarchy consisting of a controller and several sub-policies, where the controller maximizes the fair-efficient reward by switching among the sub-policies that provides diverse behaviors to interact with the environment. FEN can be trained in a fully decentralized way, making it easy to be deployed in real-world applications. Empirically, we show that FEN easily learns both fairness and efficiency and significantly outperforms baselines in a variety of multi-agent scenarios.

## 1   Introduction

Fairness is essential for human society, contributing to stability and productivity. Similarly, fairness is also the key for many multi-agent systems, *e.g.*, routing [1], traffic light control [2], and cloud computing [3]. More specifically, in routing, link bandwidth needs to be fairly allocated to packets to achieve load balance; in traffic light control, resources provided by infrastructure needs to be fairly shared by vehicles; in cloud computing, resource allocation of virtual machines has to be fair to optimize profit.

Many game-theoretic methods [4, 5, 6] have been proposed for fair division in multi-agent systems, which mainly focus on proportional fairness and envy-freeness. Most of them are in static settings, while some [7, 8, 9] consider the dynamic environment. Recently, multi-agent reinforcement learning (RL) has been successfully applied to multi-agent sequential decision-making, such as [10, 11, 12, 13, 14, 15, 16]. However, most of them try to maximize the reward of each individual agent or the shared reward among agents, without taking fairness into account. Only a few methods consider fairness, but are handcrafted for different applications [17, 18, 19], which all require domain-specific knowledge and cannot be generalized. Some methods [20, 21, 22] aim to encourage cooperation in social dilemmas but cannot guarantee fairness.

Taking fairness into multi-agent learning could help multi-agent systems become both efficient and stable. However, learning efficiency (system performance) and fairness simultaneously is a complex, multi-objective, joint-policy optimization. To tackle these difficulties, we propose a novel hierarchical RL model, FEN, to enable agents to easily and effectively learn both efficiency and fairness. First, we decompose fairness for each agent and propose *fair-efficient* reward that each agent learns its own policy to optimize it. We prove that agents achieve Pareto efficiency and fairness is guaranteed in infinite-horizon sequential decision-making if all agents maximize their own fair-efficient reward

---

[*]Corresponding author

and learn the optimal policies. However, the conflicting nature between fairness and efficiency makes it hard for a single policy to learn effectively. To overcome this, we then design a hierarchy, which consists a controller and several sub-policies. The controller maximizes the fair-efficient reward by switching among the sub-policies which directly interact with the environment. One of the sub-policies is designated to maximize the environmental reward, and other sub-policies are guided by information-theoretic reward to explore diverse possible behaviors for fairness. Additionally, average consensus, which is included in the fair-efficient reward, coordinates the policies of agents in *fully decentralized* multi-agent learning. By saying fully decentralized, we emphasize that there is no centralized controller, agents exchange information locally, and they learn and execute based on only local information.

We evaluate FEN in three classic scenarios, *i.e.*, job scheduling, the Mathew effect, and manufacturing plant. It is empirically demonstrated that FEN obtains both fairness and efficiency and significantly outperforms existing methods. By ablation studies, we confirm the hierarchy indeed helps agents to learn more easily. Benefited from distributed average consensus, FEN can learn and execute in a fully decentralized way, making it easy to be deployed in real-world applications.

## 2   Related Work

**Fairness.** There are many existing works on fair division in multi-agent systems. Most of them focus on static settings [4, 5, 6], where the information of entire resources and agents are known and fixed, while some of them work on dynamic settings [7, 8, 9], where resource availability and agents are changing. For multi-agent sequential decision-making, a regularized maximin fairness policy is proposed [8] to maximize the worst performance of agents while considering the overall performance, and the policy is computed by linear programming or game-theoretic approach. However, none of these works are learning approach. Some multi-agent RL methods [17, 18, 19] have been proposed and handcrafted for resource allocation in specific applications, such as resource allocation on multi-core systems[17], sharing network resources among UAVs[18], and balancing various resources in complex logistics networks[19]. However, all these methods require domain-specific knowledge and thus cannot be generalized. Some methods [20, 21, 22] are proposed to improve cooperation in social dilemmas. In [20], the reward is shaped for two-player Stag Hunt games, which could be agents' average reward in its multi-player version. In [20], one agent's reward is set to be the weighted average reward of two agents in two-player Stag Hunt games to induce prosociality. By extending the inequity aversion model, a shaped reward is designed in [21] to model agent's envy and guilt. A reward network is proposed in [22] to generate intrinsic reward for each agent, which is evolved based on the group's collective reward. Although cooperation in social dilemmas helps improve agents' sum reward, it does not necessarily mean the fairness is guaranteed.

The Matthew effect, summarized as the rich get richer and the poor get poorer, can be witnessed in many aspects of human society [23], as well as in multi-agent systems, such as preferential attachment in networking [24, 25] and mining process in blockchain systems [26]. The Matthew effect causes inequality in society and also performance bottleneck in multi-agent systems. Learning fairness could avoid the Matthew effect and help systems become stable and efficient.

**Multi-Agent RL.** Many multi-agent RL models have been recently proposed, such as [10, 11, 12, 13, 14, 15, 16], but all of them only consider efficiency. CommNet [11] and ATOC [13] use continuous communication for multi-agent cooperation. Opponent modeling [27, 28] learns to reason about other agents' behaviors or minds for better cooperation or competition. MADDPG [10] is designed for mixed cooperative-competitive environments. In these models, each agent only focuses on optimizing its own local reward. Thus, more capable agents will obtain more rewards and fairness is not considered. VDN [14], QMIX [15], and COMA [16] are designed for the scenario where all agents jointly maximize a shared reward. The shared reward is not directly related to fairness. Even if the shared reward is defined as the sum of local rewards of all agents, we can easily see that higher reward sum does not mean fairer.

**Hierarchical RL.** To solve more complex tasks with sparse rewards or long time horizons and to speed up the learning process, hierarchical RL trains multiple levels of policies. The higher level policies give goals or options to the lower level policies and only the lowest level applies actions to the environment. So, the higher levels are able to plan over a longer time horizon or a more complex task. Learning a decomposition of complex tasks into sub-goals are considered in [29, 30, 31], while

learning options are considered in [32, 33, 34]. However, none of these hierarchical RL models can be directly applied to learning both fairness and efficiency in multi-agent systems.

# 3 Methods

We propose **F**air-**E**fficient **N**etwork, FEN, to enable agents to learn both efficiency and fairness in multi-agent systems. Unlike existing work, we decompose fairness for each agent and propose *fair-efficient* reward, and each agent learns its own policy to optimize it. However, optimizing the two conflicting objectives is hard for a single learning policy. To this end, we propose a hierarchy specifically designed for easing this learning difficulty. The hierarchy consists a controller and several sub-policies, where the controller learns to select sub-policies and each sub-policy learns to interact with the environment in a different way. Average consensus, which is included in the fair-efficient reward, coordinates agents' policies and enables agents to learn in a fully decentralized way.

## 3.1 Fair-Efficient Reward

In the multi-agent system we consider, there are $n$ agents and limited resources in the environment. The resources are non-excludable and rivalrous (common resources), *e.g.*, CPU, memory, and network bandwidth. At each timestep, the environmental reward $r$ an agent obtains is only related to its occupied resources at that timestep. We define the utility of agent $i$ at timestep $t$ as $u_t^i = \frac{1}{t} \sum_{j=0}^{t} r_j^i$, which is the average reward over elapsed timesteps. We use the coefficient of variation (CV) of agents' utilities $\sqrt{\frac{1}{n-1} \sum_{i=1}^{n} \frac{(u^i - \bar{u})^2}{\bar{u}^2}}$ to measure fairness [35], where $\bar{u}$ is average utility of all agents. A system is said to be fairer if and only if the CV is smaller.

In multi-agent sequential decision-making, it is difficult for an individual agent to optimize the CV since it is not just related to the agent's own policy, but the joint policies of all agents. However, as the resources are limited, the upper bound of $\bar{u}$ can be easily reached by self-interested agents. Thus, $\bar{u}$ is hardly affected by an individual agent and the contribution of agent $i$ to the variance could be approximated as $(u^i - \bar{u})^2 / \bar{u}^2$. We decompose the fairness objective for each agent and propose the *fair-efficient* reward

$$\hat{r}_t^i = \frac{\bar{u}_t / c}{\epsilon + \left| u_t^i / \bar{u}_t - 1 \right|},$$

where $c$ is a constant that normalizes the numerator and is set to the maximum environmental reward the agent obtains at a timestep. In the fair-efficient reward, $\bar{u}_t / c$ can be seen as the resource utilization of the system, encouraging the agent to improve efficiency; $\left| u_t^i / \bar{u}_t - 1 \right|$ measures the agent's utility deviation from the average, and the agent will be punished no matter it is above or below the average, which leads to low variance; $\epsilon$ is a small positive number to avoid zero denominator. Each agent $i$ learns its own policy to maximize the objective $F_i = \mathbb{E} \left[ \sum_{t=0}^{\infty} \gamma^t \hat{r}_t^i \right]$, where $\gamma$ is the discount factor. The fair-efficient reward allows each agent to respond to the behaviors of other agents, which can be summarized by $\bar{u}$. Therefore, $\bar{u}$ can actually coordinate agents' policies in decentralized multi-agent learning.

**Proposition 1.** *The optimal fair-efficient policy set $\boldsymbol{\pi}^*$ is Pareto efficient in infinite-horizon sequential decision-making.*

*Proof.* We prove by contradiction. We first prove the resources must be fully occupied. Since the decision-making is infinite-horizon, the resources could be allocated in any proportion in the time domain. Assume the resources are not fully used, there must exist another $\boldsymbol{\pi}'$, under which each agent could occupy the remaining resources according to the ratio of $u^i / n\bar{u}$. Then, we have $|u^{i\prime} / \bar{u}' - 1| = |u^i / \bar{u} - 1|$, but $\bar{u}' / c > \bar{u} / c$. Thus, for each agent $i$, $F_i' > F_i$, which contradicts the pre-condition that $\boldsymbol{\pi}^*$ is optimal. We then prove $\boldsymbol{\pi}^*$ is Pareto efficient. Assume Pareto efficiency is not achieved, there must exist $\forall i, u^{i\prime} \geqslant u^i \land \exists i, u^{i\prime} > u^i$, so $\sum_{i=1}^{n} u^{i\prime} > \sum_{i=1}^{n} u^i$, which contradicts the pre-condition that the resources are fully occupied.

**Proposition 2.** *The optimal fair-efficient policy set $\boldsymbol{\pi}^*$ achieves equal allocation when the resources are fully occupied.*

*Proof.* We prove by contradiction. Assume the allocation is not equal when the resources are fully occupied, $\exists i, u^i \neq \bar{u}$. There must exist another $\boldsymbol{\pi}'$, under which agents that have $u^i > \bar{u}$ can give up

resources to make $u^i = \bar{u}$. Then, for the remaining resources and other agents, this is an isomorphic subproblem. According to *Proposition 1*, the resources will be fully occupied by other agents. After that, we have $F'_i > F_i$. This process can be repeated until $\forall i, F'_i > F_i$, which contradicts the pre-condition that $\pi^*$ is optimal.

## 3.2 Hierarchy

A learning policy could feel ambiguous while considering both fairness and efficiency since they might conflict in some states. For example, if different behaviors of other agents cause the change of $\bar{u}$, an agent may need to perform different action at a same state to maximize its fair-efficient reward. However, this is hard for a single learned policy.

To overcome this difficulty, we design a hierarchy that consists of a controller and several sub-policies parameterized respectively by $\theta$ and $\phi$, illustrated in Figure 1. The controller selects one of sub-policies by the sampled index $z_t \sim \pi_\theta(\cdot|o_t)$ based on the partial observation $o_t$. The controller receives the fair-efficient reward $\hat{r}$ and acts at a lower temporal resolution than the sub-policies. Every $T$ timesteps, the controller chooses a sub-policy and in the next $T$ timesteps, the chosen sub-policy outputs actions to interact with the environment.

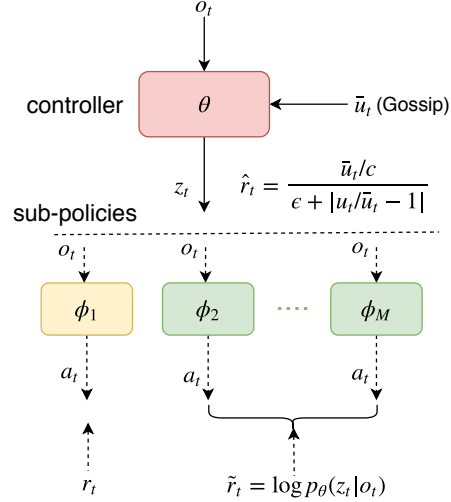

Figure 1: FEN architecture.

To obtain efficiency, we designate one of the sub-policies parameterized by $\phi_1$ to maximize the reward $r$ given by the environment. For other sub-policies, we exploit an information-theoretic objective to guide the sub-policies to explore diverse possible behaviors for fairness.

From the perspective of the controller, these sub-policies should be able to be distinguished from each other and thus the controller can have more choices. Obviously, we can not quantify the difference of sub-policies directly. However, the experienced observations under a sub-policy could indirectly reflect the policy. The more differences between sub-policies, the less the uncertainty of $z$ is, given observation $o$ under the policy. That is to say, the mutual information $I(Z; O)$ should be maximized by the sub-policy and we take it as one term of the objective. On the other hand, to explore diverse possible behaviors, the sub-policy should act as randomly as possible, so we also maximize the entropy between the action and observation $H(A|O)$. In summary, the objective of the sub-policy is to maximize

$$\begin{aligned}
J(\phi) &= I(Z;O) + H(A|O) \\
&= H(Z) - H(Z|O) + H(A|O) \\
&\approx -H(Z|O) + H(A|O) \\
&= \mathbb{E}_{z\sim\pi_\theta, o\sim\pi_\phi}[\log p(z|o)] + \mathbb{E}_{a\sim\pi_\phi}[-p(a|o)\log p(a|o)].
\end{aligned}$$

As $H(Z)$ is only related to $\theta$, $H(Z)$ can be seen a constant and be neglected. The controller just outputs the probability $p_\theta(z|o)$, and thus the first term of the objective can be interpreted as that each sub-policy tries to maximize the expected probability that it would be selected by the controller. To maximize it, we can give sub-policies a reward $\tilde{r} = \log p_\theta(z|o)$ at each timestep and use RL to train the sub-policies. The second term can be treated as an entropy regularization, which is differentiable and could be optimized by backpropagation.

The hierarchy reduces the difficulty of learning both efficiency and fairness. The controller focuses on the fair-efficient reward and learns to decide when to optimize efficiency or fairness by selecting the sub-policy, without directly interacting with the environment. Sub-policy $\phi_1$ learns to optimize the environmental reward, *i.e.*, efficiency. Other sub-policies learn diverse behaviors to meet the controller's demand of fairness. The fair-efficient reward changes slowly since it is slightly affected by immediate environmental reward sub-policy obtains. Thus, the controller can plan over a long time horizon to optimize both efficiency and fairness, while the sub-policies only optimize their own objectives within the given time interval $T$.

### 3.3 Decentralized Training

The centralized policy has an advantage in coordinating all agents' behaviors. However, the centralized policy is hard to train, facing the curse of dimensionality as the number of agents increases. FEN is a decentralized policy in both training and execution. Although each agent only focuses on its own fair-efficient reward, they are coordinated by the average consensus on utility.

In the decentralized training, each agent need to perceive the average utility $\bar{u}$ to know its current utility deviation from the average. When the number of agents is small, it is easy to collect the utility from each agent and compute the average. When the number of agents is large, it may be costly to collect the utility from each agent in real-world applications. To deal with this, we adopt a gossip algorithm for distributed average consensus [36]. Each agent $i$ maintains the average utility $\bar{u}_i$ and iteratively updates it by

$$\bar{u}_i(t+1) = \bar{u}_i(t) + \sum_{j \in \mathcal{N}_i} w_{ij} \times (\bar{u}_j(t) - \bar{u}_i(t)),$$

where $\bar{u}_i(0) = u_i$, $\mathcal{N}_i$ is the set of neighboring agents in agent $i$'s observation, and the weight $w_{ij} = 1/(\max\{d_i, d_j\} + 1)$, where the degree $d_i = |\mathcal{N}_i|$. The gossip algorithm is distributed and requires only limited communication between neighbors to estimate the average.

The training of FEN is detailed in Algorithm 1. The controller and sub-policies are trained both using PPO [37]. The controller selects one sub-policy every $T$ to interact with the environment. The selected sub-policy is updated based on the trajectory during $T$. The controller is updated based on the trajectory of every sub-policy selection and its obtained fair-efficient reward during each episode.

---

**Algorithm 1** FEN training

1: Initialize $u_i$, $\bar{u}_i$, the controller $\theta$ and sub-policies $\phi$
2: **for** episode $= 1, \ldots, \mathcal{M}$ **do**
3:     The controller chooses one sub-policy $\phi_z$
4:     **for** $t = 1, \ldots,$ max-episode-length **do**
5:         The chosen sub-policy $\phi_z$ acts to the environment and gets the reward $\begin{cases} r_t & \text{if } z = 1, \\ \log p_\theta(z|o_t) & \text{else} \end{cases}$
6:         **if** $t\%T = 0$ **then**
7:             Update $\phi_z$ using PPO
8:             Update $\bar{u}_i$ (with gossip algorithm)
9:             Calculate $\hat{r}^i = \frac{\bar{u}_i/c}{\epsilon + |u^i/\bar{u}_i - 1|}$
10:            The controller reselects one sub-policy
11:         **end if**
12:     **end for**
13:     Update $\theta$ using PPO
14: **end for**

---

## 4 Experiments

For the experiments, we design three scenarios as abstractions of job scheduling, the Matthew effect, and manufacturing plant, which are illustrated in Figure 2. We demonstrate that by each agent decentralizedly optimizing the fair-efficient reward the multi-agent system could obtain a great balance between efficiency and fairness, and that the hierarchy indeed helps to learn both fairness and efficiency more easily. In the experiments, we compare FEN against several baselines which have different optimization objectives and are summarized as follows.

- `Independent` agents are fully self-interested and each agent maximizes its expected sum of discounted environmental rewards $\psi_i = \mathbb{E}\left[\sum_{t=0}^{\infty} \gamma^t r_t^i\right]$.

- `Inequity Aversion` agents receive a shaped reward $r_i - \frac{\alpha}{N-1} \sum \max(r_j - r_i, 0) - \frac{\beta}{N-1} \sum \max(r_i - r_j, 0)$ to model the envy and guilt [21].

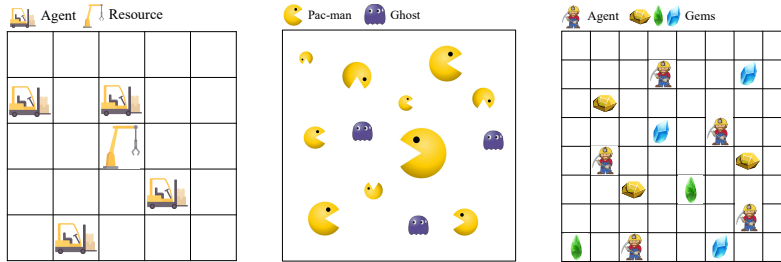

Figure 2: Illustration of experimental scenarios: job scheduling (*left*), the Matthew effect (*mid*), manufacturing plant (*right*).

- `Avg` agents take the average reward of agents as a shared reward and maximize $\mathrm{avg}\psi = \sum \psi_i/n$ [20].

- `Min` agents consider the worst performance of agents and maximize $\mathrm{min}\psi$.

- `Min+`$\alpha$`Avg` agents consider both the worst performance and system performance and maximize the regularized maximin fairness [8], $\mathrm{min}\psi + \alpha\mathrm{avg}\psi$.

Note that in the last three baselines, all agents share the same optimization objective. To ensure the comparison is fair, the basic hyperparameters are all the same for FEN and the baselines, which are summarized in Appendix. The details about the experimental setting of each scenario are also available in Appendix. Moreover, the code of FEN is at https://github.com/PKU-AI-Edge/FEN.

## 4.1 Job Scheduling

In this scenario of job scheduling, we investigate whether agents can learn to fairly and efficiently share a resource. In a $5 \times 5$ grid world, there are $4$ agents and $1$ resource, illustrated in Figure 2 (*left*). The resource's location is randomly initialized in different episodes, but fixed during an episode. Each agent has a local observation that contains a square view with $3 \times 3$ grids centered at the agent itself. At each timestep, each agent can move to one of four neighboring grids or stay at current grid. If the agent occupies the resource (move to or stay at the resource's location), it receives a reward of $1$, which could be seen as the job is scheduled, otherwise the reward is $0$. Two agents cannot stay at a same grid, making sure the resource can only be occupied by one agent at a timestep. We trained all the methods for five runs with different random seeds. All experimental results are presented with standard deviation (also in other two scenarios). Moreover, as all agents are homogeneous in the task, we let agents share weights for all the methods.

Table 1 shows the performance of FEN and the baselines in terms of resource utilization (sum of utility), coefficient of variation (CV) of utility (fairness), min utility, and max utility. `Independent` has the highest resource utilization, but also the worst CV. Self-interested `Independent` agents would not give up the resource for fairness, which is also witnessed by that min utility is $0$ and max utility is $0.88$, close to resource utilization. FEN has the lowest CV and the highest min utility. As only one agent can use the resource at a time, fairly sharing the resource among agents inevitably incurs the reduction of resource utilization. However, FEN can obtain much better fairness at a subtle cost of resource utilization, and its resource utilization is slightly less than `Independent`. Maximizing $\mathrm{avg}\psi$ causes high CV since the average is not directly related to fairness. Its resource utilization is also lower, because $\mathrm{avg}\psi$ is determined by all the agents, making it hard for individual agents to optimize by decentralized training. For the same reason, $\mathrm{min}\psi$ is hard to be maximized by individual agents. The regularized maximin fairness reward $\mathrm{min}\psi + \alpha\mathrm{avg}\psi$ is designed to obtain a balance between fairness and resource utilization. However, due to the limitations of these two objective terms, `Min+`$\alpha$`Avg` is much worse than FEN. The CV of `Inequity Aversion` is better than `Independent` but still worse than FEN, and the resource utilization is much lower, showing modeling envy and guilt is not effective in fairness problems. Moreover, the hyperparameters $\alpha$ and $\beta$ might greatly affect the performance.

Table 1: Job scheduling

|  |  | resource utilization | CV | min utility | max utility |
|---|---|---|---|---|---|
| Independent | | $96\% \pm 11\%$ | $1.57 \pm 0.26$ | $0$ | $0.88 \pm 0.17$ |
| Inequity Aversion | | $72\% \pm 9\%$ | $0.69 \pm 0.17$ | $0.04 \pm 0.01$ | $0.35 \pm 0.12$ |
| Min | | $47\% \pm 8\%$ | $0.30 \pm 0.07$ | $0.07 \pm 0.02$ | $0.16 \pm 0.05$ |
| Avg | | $84\% \pm 7\%$ | $0.75 \pm 0.13$ | $0.05 \pm 0.03$ | $0.46 \pm 0.17$ |
| Min+$\alpha$Avg | | $63\% \pm 5\%$ | $0.39 \pm 0.03$ | $0.09 \pm 0.03$ | $0.24 \pm 0.06$ |
| FEN | | $\mathbf{90\%} \pm 5\%$ | $\mathbf{0.17} \pm 0.05$ | $\mathbf{0.18} \pm 0.03$ | $0.28 \pm 0.07$ |
| FEN w/o Hierarchy | | $57\% \pm 13\%$ | $0.22 \pm 0.06$ | $0.10 \pm 0.03$ | $0.18 \pm 0.11$ |
| centralized policy | Min | $12\% \pm 4\%$ | $0.82 \pm 0.11$ | $0$ | $0.06 \pm 0.03$ |
| | Avg | $61\% \pm 5\%$ | $1.46 \pm 0.14$ | $0$ | $0.53 \pm 0.06$ |
| | Min+$\alpha$Avg | $19\% \pm 5\%$ | $0.57 \pm 0.05$ | $0.02 \pm 0.01$ | $0.09 \pm 0.03$ |
| w/ Hierarchy | Min | $62\% \pm 9\%$ | $0.31 \pm 0.11$ | $0.09 \pm 0.02$ | $0.21 \pm 0.05$ |
| | Avg | $84\% \pm 6\%$ | $0.61 \pm 0.14$ | $0.08 \pm 0.03$ | $0.41 \pm 0.07$ |
| | Min+$\alpha$Avg | $71\% \pm 8\%$ | $0.28 \pm 0.09$ | $0.11 \pm 0.04$ | $0.26 \pm 0.06$ |

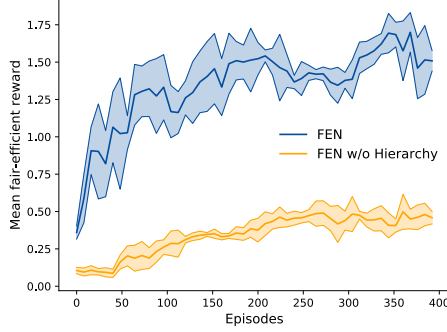

Figure 3: Learning curves of FEN and FEN w/o Hierarchy in job scheduling.

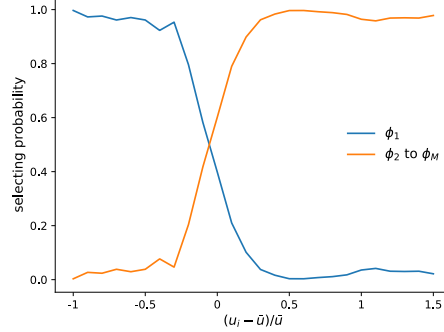

Figure 4: Probability of selecting different sub-policies in terms of $(u_i - \bar{u})/\bar{u}$.

Since $\min\psi$, $\mathrm{avg}\psi$, and $\min\psi + \alpha\mathrm{avg}\psi$ do not depend on individual agents, but all agents, we adopt a centralized policy, which takes all observations and outputs actions for all agents, to optimize each objective. As shown in Table 1, the centralized policies for Min, Avg, and Min+$\alpha$Avg are even worse than their decentralized versions. Although the centralized policy could coordinate agents' behaviors, it is hard to train because of the curse of dimensionality. We also tried the centralized policy in the Matthew effect and manufacturing plant, but it did not work and thus is omitted.

*Does the hierarchy indeed help the learning of FEN?* To verify the effect of the hierarchy, we trained a single policy to maximize the fair-efficient reward directly without the hierarchy. Figure 3 illustrates the learning curves of FEN and FEN w/o Hierarchy, where we can see that FEN converges to a much higher mean fair-efficient reward than FEN w/o Hierarchy. As shown in Table 1, although FEN w/o Hierarchy is fairer than other baselines, the resource utilization is mediocre. This is because it is hard for a single policy to learn efficiency from the fair-efficient reward. However, in FEN, one of the sub-policies explicitly optimizes the environmental reward to improve the efficiency, other sub-policies learn diverse fairness behaviors, and the controller optimizes fair-efficient reward by long time horizon planing. The hierarchy successfully decomposes the complex objective and reduce the learning difficulty.

To further verify the effectiveness of the hierarchy, we use the hierarchy with other baselines. The controller maximizes each own objective and the sub-policies are the same as FEN. Table 1 shows their performance has a certain degree of improvement, especially the resource utilizations of Min and Min+$\alpha$Avg raise greatly and the CV of Min+$\alpha$Avg reduces significantly. That demonstrates the hierarchy we proposed could reduce learning difficulty in many general cases with both global and local objectives. However, these baselines with the hierarchy are still worse than FEN in both resource utilization and CV, verifying the effectiveness of the fair-efficient reward.

In order to analyze the behavior of the controller, in Figure 4 we visualize the probability of selecting sub-policy $\phi_1$ and other sub-policies in terms of the utility deviation from average, $(u_i - \bar{u})/\bar{u}$. It shows when the agent's utility is below average, the controller is more likely to select $\phi_1$ to occupy the resources, and when the agent's utility is above average, the controller tends to select other sub-policies to improve fairness. The controller learns the sensible strategy based on the fair-efficient reward.

## 4.2 The Matthew Effect

In this scenario of the Matthew effect, we investigate whether agents can learn to mitigate/avoid the Matthew effect. In the scenario, there are 10 pac-men (agents) initialized with different positions, sizes, and speeds and also 3 stationary ghosts initialized at random locations, illustrated in Figure 2 (*mid*). Each pac-man can observe the nearest three other pac-men and the nearest one ghost. It could move to one of four directions or stay at current position. Once the distance between the pac-men and a ghost is less than the agent's size, the ghost is consumed and the agent gets a reward 1. Then, a new ghost will be generated at a random location. When the agent gets a reward, its size and speed will increase correspondingly until the upper bounds are reached. In this scenario, the pac-man who consumes more ghosts becomes larger and faster, making consume ghosts easier. So, there exists inherent inequality in the setting. We trained all the models for five runs with different random seeds. As pac-men are homogeneous, we let pac-men share weights for all the methods.

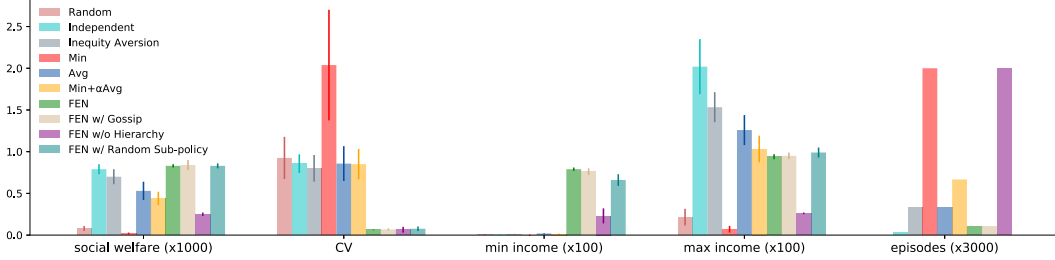

Figure 5: The Matthew effect

Figure 5 shows the performance of FEN and the baselines in terms of social welfare (total ghosts consumed by all the pac-men), CV, and min and max income (consumed ghosts) among pac-men, episodes to converge. The detailed results are available in Appendix. Random pac-men take random actions and their CV shows the inherent unfairness of this scenario. Min pac-men cannot learn reasonable policies, because $\min\psi$ is always closed to $0$. Min+$\alpha$Avg is only a little fairer than Avg since the effect of $\min\psi$ is very weak. Independent causes the Matthew effect as indicated by the min (close to $0$) and max (more than $200$) income, where pac-man with initial larger size becomes larger and larger and ghosts are mostly consumed by these larger pac-men. Inequity Aversion is slightly fairer than Independent but lower social welfare.

Although Independent has the largest pac-man which consumes ghosts faster than others, this does not necessarily mean they together consume ghosts fast. FEN is not only fairer than the baselines but also has the highest social welfare, even higher than Independent. FEN pac-men have similar sizes and consume more ghosts than the baselines. This demonstrates FEN is capable of tackling the Matthew effect and helps social welfare increase. FEN w/o Hierarchy focuses more on fairness, neglecting the efficiency as in the scenario of job scheduling. Moreover, learning without hierarchy is much slower than FEN in this scenario, as illustrated in Figure 6. FEN w/o Hierarchy takes about $6000$ episodes, while FEN takes only about $300$ episodes, confirming that the hierarchy indeed speeds up the training.

*Does distributed average consensus affect the performance of FEN?* Instead of using the centrally computed average utility, we employ the gossip algorithm to estimate the average utility, where each agent only exchanges information with the agents in its observation. As shown in Figure 5, FEN w/ Gossip performs equivalently to FEN with only slight variation on each performance metric. The learning curve of FEN w/ Gossip is also similar to FEN, as illustrated in Figure 6. These confirm that FEN can be trained in a fully decentralized way.

*Do sub-policies really learn something useful?* To answer this question, after the training of FEN, we keep the learned weights $\theta$ and $\phi_1$ and replace other sub-policies with a random sub-policy. Once the controller chooses other sub-policies instead of $\phi_1$, the agent will perform

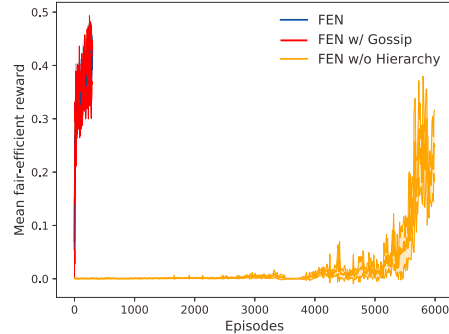

Figure 6: Learning curves of FEN, FEN w/ Gossip, and FEN w/o Hierarchy in the Matthew effect.

random actions. In this FEN w/ random sub-policy, the min income become lower than FEN and CV becomes higher, because the random sub-policy cannot provide fairness behavior the controller requests. To investigate the difference of learned sub-policies and random sub-policy, we fix the three ghosts as a triangle at the center of the field and visualize the distribution of an agent's positions under each sub-policy, as illustrated in Figure 7. It is clear that the learned sub-policies keep away from the three ghosts for fairness and their distributions are distinct, concentrated at different corners, verifying the effect of the information-theoretic reward.

### 4.3 Manufacturing Plant

In this scenario of manufacturing plant, we investigate whether agents with different needs can learn to share different types of resources and increase the production in a manufacturing plant. In a $8 \times 8$

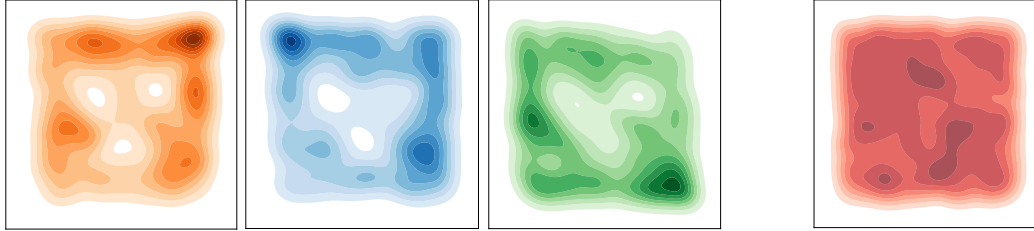

Figure 7: Visualization of learned sub-policies (*left three*) and random sub-policy (*right*) in the Matthew effect.

grid world, there are $5$ agents and $8$ gems, as illustrated in Figure 2 (*right*). The gems belong to three types (y, g, b). Each agent has a local observation that contains a square view with $5 \times 5$ grids centered at the agent itself, and could move to one of four neighboring grids or stay. When an agent moves to the grid of one gem, the agent collects the gem and gets a reward of $0.01$, and then a new gem with random type and random location is generated. The total number of gems is limited, and when all the gems are collected by the agents, the game ends. Each agent has a unique requirement of numbers for the three types of gems to manufacture a unique part of the product and receive a reward $1$. Each product is assembled by the five unique parts manufactured by the five agents, respectively. So, the number of manufactured products is determined by the least part production among the agents. Due to the heterogeneity, we let each agent learn its own weights for FEN and the baselines.

Table 2 shows the performance of FEN and the baselines in terms of resource utilization (the ratio of the number of gems consumed to manufacture the products over the total number of gems), CV, number of products (minimum number of manufactured parts among agents), and max number of manufactured parts among agents. In this scenario, agents need to learn to collect the right gems and then to balance the parts manufactured by each agent (*i.e.*, manufacturing similar large number of parts), because the unused collected gems and redundant parts will be wasted. FEN manufactures the most products, more than two times than the baselines. The more products are assembled, the higher the resource utilization is. Thus, FEN also has the highest resource utilization. Moreover, FEN is also the fairest one. Although FEN w/o Hierarchy agents are fairer than other baselines, they all manufacture less parts and hence eventually less products. Avg agents assemble the least products, though one agent manufactures the largest number of parts, resulting in serious waste.

Table 2: Manufacturing plant

|  | resource utilization | CV | no. products | max parts |
|---|---|---|---|---|
| Independent | $28\% \pm 5\%$ | $0.38 \pm 0.08$ | $19 \pm 3$ | $58 \pm 8$ |
| Inequity Aversion | $27\% \pm 6\%$ | $0.27 \pm 0.06$ | $19 \pm 4$ | $42 \pm 7$ |
| Min | $29\% \pm 6\%$ | $0.26 \pm 0.01$ | $20 \pm 4$ | $41 \pm 7$ |
| Avg | $13\% \pm 3\%$ | $0.63 \pm 0.07$ | $9 \pm 2$ | $71 \pm 9$ |
| Min+$\alpha$Avg | $34\% \pm 6\%$ | $0.28 \pm 0.01$ | $23 \pm 4$ | $45 \pm 7$ |
| FEN | $\mathbf{82\%} \pm 5\%$ | $\mathbf{0.10} \pm 0.03$ | $\mathbf{48} \pm 3$ | $63 \pm 3$ |
| FEN w/o Hierarchy | $22\% \pm 3\%$ | $0.18 \pm 0.07$ | $15 \pm 1$ | $24 \pm 4$ |

## 5 Conclusion

We have proposed FEN, a novel hierarchical reinforcement learning model to learn both fairness and efficiency, driven by fair-efficient reward, in multi-agent systems. FEN consists of one controller and several sub-policies, where the controller learns to optimize the fair-efficient reward, one sub-policy learns to optimize the environmental reward, and other sub-policies learn to provide diverse fairness behaviors guided by the derived information-theoretic reward. FEN can learn and execute in a fully decentralized way, coordinated by average consensus. It is empirically demonstrated that FEN easily learns both fairness and efficiency and significantly outperforms baselines in a variety of multi-agent scenarios including job scheduling, the Matthew effect, and manufacturing plant.

**Acknowledgments**

This work was supported in part by NSF China under grant 61872009, Huawei Noah's Ark Lab, and Peng Cheng Lab.

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
