[Supplementary Material]

# Appendix

## Hyperparameters

In the experiments, we use PPO for every RL agent. The PPO structure keeps same for the controller and sub-policies of FEN, and also for the baselines. The value network and policy network are MLPs with two 256-unit hidden layers and ReLU activation. The learning rates for the value network and policy network of PPO are $10^{-3}$ and $3 \times 10^{-4}$, respectively. We trained all networks using Adam optimizer. The discounted factor is $\gamma = 0.98$. For Min+$\alpha$Avg agents, $\alpha = 0.01$. For Inequity Aversion agents, $\alpha = 5$ and $\beta = 0.05$, the setting used in [21]. For FEN, the number of the sub-policies is 4, $\epsilon$ in the fair-efficient reward is set to 0.1. $T$ is 25, 50, and 50 in job scheduling, the Matthew effect, and manufacturing plant, respectively.

## Experimental Settings

In the scenario of job scheduling, we trained all the models for five runs with different random seeds, each episode contains 1000 timesteps.

In the scenario of the Matthew effect, the size and speed of pac-man are randomly initiated between $(0.01, 0.04)$ and $(0.018, 0.042)$, respectively. Each time after a pac-men consumes a ghost, its size will increase by 0.005 and the speed will increase by 0.004. The max size is 0.15 and the max speed is 0.13. We trained all the models for five runs with different random seeds, where each episode contains 1000 timesteps.

In the scenario of manufacturing plant, the total number of gems is 700. The unique requirements of numbers of three types of gems $(L_y, L_g, L_b)$ are $(2, 1, 0), (1, 0, 1), (0, 1, 1), (1, 1, 0), (0, 1, 2)$, respectively, for the five agents. We trained all the models for five runs with different random seeds, where each episode ends after all the gems are collected by agents.

All the experiments were conducted on Dell XPS desktops with i7-8700k 6-Core processors and 1080TI GPUs.

## Experimental Results

The details of the experimental results in the Matthew effect are shown in Table 3.

Table 3: The Matthew effect

|  | social welfare | CV | min income | max income | episodes |
|---|---|---|---|---|---|
| Random | $84 \pm 30$ | $0.93 \pm 0.25$ | 1 | $22 \pm 10$ | - |
| Independent | $791 \pm 62$ | $0.86 \pm 0.11$ | 1 | $202 \pm 33$ | 100 |
| Inequity Aversion | $702 \pm 90$ | $0.80 \pm 0.16$ | 2 | $152 \pm 18$ | 1000 |
| Min | $18 \pm 8$ | $2.04 \pm 0.66$ | 0 | $7 \pm 4$ | 6000 |
| Avg | $527 \pm 113$ | $0.86 \pm 0.21$ | 2 | $126 \pm 18$ | 1000 |
| Min+$\alpha$Avg | $441 \pm 75$ | $0.85 \pm 0.18$ | $1 \pm 1$ | $103 \pm 16$ | 2000 |
| FEN | $\mathbf{830} \pm 22$ | $\mathbf{0.06} \pm 0.01$ | $\mathbf{79} \pm 2$ | $94 \pm 3$ | 300 |
| FEN w/ Gossip | $\mathbf{841} \pm 55$ | $\mathbf{0.07} \pm 0.01$ | $\mathbf{76} \pm 4$ | $95 \pm 4$ | 300 |
| FEN w/o Hierarchy | $251 \pm 12$ | $0.06 \pm 0.04$ | $23 \pm 2$ | $26 \pm 1$ | 6000 |
| FEN w/ Random Sub-policy | $834 \pm 47$ | $0.08 \pm 0.02$ | $66 \pm 7$ | $99 \pm 6$ | - |