[Reviews · NeurIPS 2019]

Reviewer 1



The authors propose a Fair-Efficient Network to better to train decentralized multi-agent reinforcement learning systems in tasks that involve resource allocation. In particular they introduce a shaping reward and a hierarchical model which they train with PPO on three new reinforcement learning environments (the code of which is made available). Their model outperforms several baselines, and ablation studies demonstrate the usefulness of the hierarchical nature of the model. The aims of the work are clear and well-stated. However, there are significant omissions in the review of related literature. Several papers have studied fairness in the context of common resources in multi-agent reinforcement learning prior to this work, namely: https://arxiv.org/abs/1803.08884 https://arxiv.org/abs/1811.05931 and related works cited therein. Although this impacts on the originality of the paper, the methods used to generate fairness are different in this work, so with an improved literature review and drawing contrasts to the previous work, the paper could be greatly strengthened. However, the choice of fair-efficient reward appears fairly arbitrary in this work. The equation in 3.1 could be replaced by many other options which also satisfied the criteria of Propositions 1 and 2. This is a weakness of the work, and the authors would do well to present a cogent argument for the functional form chosen. The hierarchical model is the greatest strength of the paper. The authors derive an interesting information-theoretic optimization goal for the sub-policies. The results in Figure 6 are particularly striking. Indeed, it would be interesting to see whether merely using the hierarchical model in conjunction with some of the other baselines obviates the need for the fair-efficient reward structure. Comments / experiments in this direction would strengthen the paper. In general there are some infelicities in wording which could be ameliorated on a proof-read. Moreover, the first two pages are fairly repetitive and could be condensed. On the other hand the descriptions of experiments are clear and concise. More details of hyperparameters and seeds chosen for the reinforcement learning training and models should be provided before publication for the purpose of reproducibility. Error bars and confidence intervals are provided in the results, but currently without sufficient explanation. === Response to authors: I was impressed by the response of the authors. They have clearly taken into account the feedback of the reviewers and make cogent arguments for the benefits of their method. They have also provided comparison against prior work and demonstrated the improvements that their work can bring. Moreover, it is now much clearer to me how both the hierarchy and the intrinsic motivation are beneficial and indeed complementary. Therefore I have increased my score by 2 points, and argue for the acceptance of this paper.

Reviewer 2



Summary: The authors propose a novel HRL algorithm (named FEN) for training fair and efficient policies in MARL settings. They design a new type of reward that takes both efficiency and fairness into consideration. FEN contains one meta controller and a few sub-policies. The controller learns to optimize the fair and efficient reward while one sub-policy is trained to optimize external reward (from the environment) and other sub-policies provide diverse but fair behavior. They show their method learns both fair and efficient behavior at the same time and outperforms relevant baselines on 3 different tasks: job scheduling, the Matthew effect, and manufacturing plant. They also show that the agents achieve Pareto efficiency and fairness is guaranteed in infinite-horizon sequential decision making if all agents play optimal policies (that maximize their own fair=efficient reward). Strengths: - the paper is generally clear and well-structured - the proposed approach is novel as far as I know - I believe the authors address an important topic in multi-agent reinforcement learning: designing agents that are both efficient and fair at the same time. I think people in the community would be interested in this work and could build on top of it. - Their method has some theoretical guarantees, which makes it quite appealing. It is also grounded in game theory aspects. - The approach is thoroughly evaluated on 3 different tasks and shows significant gains in fairness without losing in overall performance. - The authors do ablation studies to emphasize the importance of using a hierarchical model and also the effectiveness of the Gossip version of the model Weaknesses: - the paper is missing some references to other papers addressing fairness in MARL such as Hughes et al. 2018, Freire et al. 2019, and other related work on prosocial artificial agents such as Peysakhovich et al. 2018 etc. - the paper could benefit from comparisons against other baselines using fairness and prosocial ideas such as the ones proposed by Hughes et al. 2018 and Peysakhovich et al 2018 - I've find the use of "decentralized training" to not be entirely correct given that the agents need access to the utility of all (or at least some, in the Gossip version) agents in order to compute their own rewards. this is generally private information and, so I wouldn't consider it fully decentralized. while the Gossipy version of the model that only uses the utilities of neighboring agents helps to relax some of these constraints, the question of how these rewards can be obtains in a setting with truly independent learners remains. Please clarify if there is a misunderstanding on my part. Other Comments: - it wasn't very clear to me from the paper what happens at test time. is it the same as during training in that the meta-controller picks one of the policies to act? - it would be interesting to look at the behavior of the controller for gaining a better understanding of when it decides to pick the sub-policy that maximizes external reward and when it picks the sub-policies that maximize the fairness reward. - in particular, it would be valuable how the different types of policies are balanced and what factors influence the trade-off between the sub-policy maximizing external reward and those with diverse fair behavior (i.e. current reward, observation, training stage, environment etc.)

Reviewer 3



This paper assumes that the reward of each agent is independent from each other and the overall reward is addictive. This limits its applicability to more general multi-agent systems where multiple agents share the reward with common goals. This work uses the coefficient of variation (CV) of agents’ utilities to measure fairness. However, it is not clear how such fairness measurement is achieved with the proposed fair-efficient reward. There should be a formal proof that the CV is minimized given the decomposition of the reward. Propositions 1 and 2 are uninformative. It is not clear why the switch of sub-policies is necessary. As stated in the beginning of Section 3.2, if other agents change their behaviors, an agent may need to perform different action at the same sate to maximize its fair-efficient reward. This seems incorrect because all the agents are trained by the same algorithm so they must be coordinated. It is true that this is a multi-objective optimization problem with fairness and efficiency. However, the agents should make their choice eventually to balance the fairness and efficiency. So these two objectives can be combined and the controller is unnecessary. The decentralized training is only for one agent. It is not clear how the agents can coordinate with the policies trained in a decentralized fashion to achieve fairness.

[Author Response · NeurIPS 2019]

We gratefully appreciate the efforts made by all the reviewers. Thanks to **Reviewers #1** and **#2** for bringing up the missing references. Hughes et al. [2018] extend the inequity aversion model and define a shaped reward $r_i - \frac{\alpha}{N-1} \sum \max(r_j - r_i, 0) - \frac{\beta}{N-1} \sum \max(r_i - r_j, 0)$. Wang et al. [2018] design a reward network generating intrinsic reward, evolved based on the group's collective reward. Peysakhovich et al. [2018] propose a shaped reward $r_i = \alpha r_i + (1 - \alpha) r_j$ for two-player Stag Hunts. The baseline `Avg` can be seen as its multi-player version as the authors claim. These works aim to improve cooperation but cannot guarantee fairness. We compare against Hughes et al. [2018], `Inequity Aversion`, in job scheduling. Table 1 shows the CV of `Inequity Aversion` is better than `Independent` but still much worse than `FEN`, and the resource utilization is much lower. That shows `Inequity Aversion` cannot solve job scheduling fairly and efficiently. More details will be included in the final version. We will also include the review of the missing references in the final version.

Table 1: Job scheduling

| | resource utilization | CV |
|---|---|---|
| Independent | 96% ±11% | 1.57 ±0.26 |
| FEN | **90%** ±5% | **0.17** ±0.05 |
| Inequity Aversion | 72% ±9% | 0.69 ±0.17 |

Table 2: Hierarchy

| | resource utilization | CV |
|---|---|---|
| Min w/ Hierarchy | 62% ±9% | 0.31 ±0.11 |
| Avg w/ Hierarchy | 84% ±6% | 0.61 ±0.14 |
| Min+$\alpha$Avg w/ Hierarchy | 71% ±8% | 0.28 ±0.09 |
| FEN | **90%** ±5% | **0.17** ±0.05 |

**Reviewer #1** To verify the effectiveness of the hierarchy, we use the hierarchy with other baselines in job scheduling. Table 2 shows their performance has a certain degree of improvement, especially the resource utilizations of `Min` and `Min+`$\alpha$`Avg` raise greatly and the CV of `Min+`$\alpha$`Avg` reduces significantly. That demonstrates the effect of the hierarchy. However, these baselines with the hierarchy are still worse than `FEN` in both resource utilization and CV, verifying the effectiveness of the fair-efficient reward.

The intuition of the fair-efficient reward is to maximize the resource utilization while punish the agent's utility deviation from the average, taking both fairness and efficiency into consideration. Also, the fair-efficient reward is suitable for decentralized training, which can be easily coordinated by $\bar{u}$. We design the fair-efficient reward, prove it satisfies the criteria of Propositions 1 and 2, and empirically verify it really works well.

The main hyperparameters are contained in the Appendix, we will make a further supplement in the final version. All the results are obtained by five runs with different random seeds and presented with standard deviation (line 234), and we will make it clearer.

**Reviewer #2** It is really a constructive suggestion to analyze the behavior of the controller. Figure 1 visualizes the probability of selecting sub-policy $\phi_1$ and other sub-policies in terms of the utility deviation from average, $(u_i - \bar{u})/\bar{u}$. It shows when the agent's utility is below average, the controller is more likely to select $\phi_1$ to occupy the resources, and when the agent's utility is above average, the controller tends to select other sub-policies to improve fairness. Thus, it can be seen that the balance of these two kinds of sub-policies depends on the current fair-efficient reward.

Sorry for the confusion induced by "decentralized training." By that we mean the training of each agent requires only limited information exchanging with neighboring agents. We will use more precise expression to replace that in the final version. At test time, it is the same as training where the controller chooses one sub-policy every $T$ timesteps.

Figure 1: Selecting probability over $(u_i - \bar{u})/\bar{u}$.

**Reviewer #3** We explain the necessity of the hierarchy from three aspects. First, the hierarchy reduces the difficulty of learning both efficiency and fairness. Since the problem is a multi-objective optimization, the learning difficulty for a single neural network cannot be neglected. In the hierarchy, each sub-policy focuses on its own easy objective and there is no conflict; the controller focuses on the fair-efficient reward by selecting the sub-policies, without directly interacting with the environment. Second, the fair-efficient reward changes slowly since it is slightly affected by the sub-policy's action in one timestep. Thus, the controller can plan over a long-time horizon to optimize both objectives. Third, in all the three experiments, FEN with hierarchy outperforms the version without hierarchy, verifying the hierarchy helps the learning greatly. We also use the hierarchy with other baselines and their performance improves as shown in Table 2, which also shows the effectiveness of the hierarchy.

Although the training is decentralized, the agent can obtain the average utility $\bar{u}$ by Gossip, and hence each agent knows the utility deviation from the average. To make its own fair-efficient reward higher, the agent with lower $u_i$ must occupy more resources and the agent with higher $u_i$ than $\bar{u}$ will choose to not occupy resources. That is why the policies can be coordinated. Each agent only focuses on its own fair-efficient reward and the fairness could be achieved.

This paper focuses on achieving both fairness and efficiency in the context of *common resources* which is one of important fields in MARL. Propositions 1 and 2 have proved that the resources will be *fully* occupied and *equally* allocated *in infinite-horizon sequential decision-making*, thus the CV is also minimized according to its definition.

[Meta-Review · NeurIPS 2019]

There was general consensus amongst the reviewers that this paper is well written and presents some interesting and novel ideas w.r.t. fairness in MAS. There were quite some concerns though initially w.r.t. related work, advantages and the technical aspects of the proposed approach. The rebuttal has brought a lot of clarity w.r.t all those identified issues which has lead to general agreement in the discussion of the paper that this work is worthy of publication at NeurIPS. It’s important though that the authors do take care of including the promised missing details and extended description of related work in the crc of their article.